# Peatland Hydrological Dynamics as A Driver of Landscape Connectivity and Fire Activity in the Boreal Plain of Canada

**Dan K. Thompson** [1,2,*] **, Brian N. Simpson** [1] **, Ellen Whitman** [1,2] **, Quinn E. Barber** [1] **and Marc-André Parisien** [1]

1   Natural Resources Canada, Canadian Forest Service, Northern Forestry Centre,
    Edmonton, AB T6H 3S5, Canada
2   Canadian Partnership for Wildland Fire Science, Department of Renewable Resources, University of Alberta,
    Edmonton, AB T6G 2H1, Canada
*   Correspondence: daniel.thompson@canada.ca

**Abstract:** Drought is usually the precursor to large wildfires in northwestern boreal Canada, a region with both large wildfire potential and extensive peatland cover. Fire is a contagious process, and given weather conducive to burning, wildfires may be naturally limited by the connectivity of fuels and the connectivity of landscapes such as peatlands. Boreal peatlands fragment landscapes when wet and connect them when dry. The aim of this paper is to construct a framework by which the hydrological dynamics of boreal peatlands can be incorporated into standard wildfire likelihood models, in this case the Canadian Burn-P3 model. We computed hydrologically dynamic vegetation cover for peatlands (37% of the study area) on a real landscape in the Canadian boreal plain, corresponding to varying water table levels representing wet, moderate, and severely dry fuel moisture and hydrological conditions. Despite constant atmospheric drivers of fire spread (air temperature, humidity, and wind speed) between drought scenarios, fire activity increased 6-fold in moderate drought relative to a low drought baseline; severe (1 in 40 years) drought scenarios drove fires into previously fire-restrictive environments. Fire size increased 5-fold during moderate drought conditions and a further 20–25% during severe drought. Future climate change is projected to lead to an increase in the incidence of severe drought in boreal forests, leading to increases in burned area due to increasing fire frequency and size where peatlands are most abundant. Future climate change in regions where peatlands have historically acted as important barriers to fire spread may amplify ongoing increases in fire activity already observed in Western North American forests.

**Keywords:** wildfire; peatlands; boreal

## 1. Introduction

In boreal North America, peatlands (wetlands with deep organic soils) cover 20% of land area, far more abundant than the 2.8% of global land area occupied by peat [1]. Boreal peatlands range from semi-aquatic treeless grass and sedge systems to closed-canopy forests, with all peatlands sharing a minimum 40 cm thick organic soil layer composed primarily of decomposed moss [2]. The limited silvicultural value of peatlands and their perception as being perennially saturated at the surface has led to the exclusion of peatlands from the current wildfire behavior models in Canada, though attention has been paid to the role of forested peatlands both in carbon emissions [3], as well as their fire ecology [4], ecohydrology [5], and fuel structure [6,7]. Peatlands in North America are primarily disturbed by wildfire [8], with the regions of highest wildfire occurrence [9] often overlapping with the

highest peatland cover [7]. Spatially, upland and lowland areas in the boreal region of Canada may only be 1 or 2 meters of elevation apart, and heavily intermixed at all spatial scales ([10]; Figure S1).

Though increased precipitation is forecast under a changing climate in many boreal and subarctic regions of North America, increases in air temperature and evaporative demand will likely yield a net decrease in fuel moisture and ecosystem water availability [11,12]. For upland boreal forests, where flammable forest floor vegetation is limited to a thin layer of undecomposed or partially decomposed material, fuel moisture is largely depleted after two weeks without precipitation [13]. However, in deep organic systems such as peatlands, this effect is more complex, owing to the connection of surface fuel conditions with the water table and numerous natural negative ecohydrological feedbacks that strive to maintain a high water table [14]. When the ecohydrological feedbacks are overwhelmed by prolonged drying, widespread burning of peatlands can result [15].

Peatland water loss and drying dynamics differ from uplands both in time scale and the location of water storage. Within a peatland ecosystem, the decline of the water table beneath the lowermost levels of the undulating peat surface allows for the increasing drying of tree and shrub litter on the forest floor, as well as a gradual drying of any peat above the water table itself. This response of boreal peatland water tables to increasing water deficit can be approximated by the Drought Code values from the Canadian Fire Weather Index (FWI) System [16]. While this relationship has been demonstrated at numerous individual long-term monitoring sites, there is currently no clear path to implement these hydrological relationships within a fire danger and risk framework. As a result, the manner in which peatland drying impacts fire regimes is largely unexplored.

In this paper, we propose an empirical framework of fuel mapping and fire weather analysis in order to dynamically modify wildfire fuels maps to account for hydrological tipping points in boreal peatland flammability. Specifically, this paper presents a framework that: (i) predicts water-table depth in differing wetland types using a simple weather-driven approach; (ii) determines the hydrological thresholds in peatlands that correspond to markedly higher wildfire spread rates; (iii) alters ignition patterns, fire spread rates, and ultimately landscape fire likelihood. We do not consider the spatial patterns in human or natural values across the landscape that in turn impact fire risk (defined as the intersection of likelihood and effects [17]). Using a Boreal plains landscape of northwestern Canada with a largely natural fire regime as a case study, we show how the quantification of wildfire activity is responsive to the dynamic assessment of wetland hydrology. The study aims to demonstrate the impact of increased fuel continuity due to drought that promotes the large size and high frequency of fires across a wetland-abundant landscape.

## 2. Materials and Methods

### 2.1. Study Area

The study area (5.1 Mha) lies in the transition between the Canadian Boreal plains and Taiga plains ecozones south of Great Slave Lake to the Peace-Athabasca Delta and Lake Claire, spanning lands of the Northwest Territories, the province of Alberta, and much of Wood Buffalo National Park (Figure 1). The Boreal and Taiga plains ecozones are a complex arrangement of deep glacial and lacustrine sedimentary deposits that facilitate deep groundwater inputs in fens and provide poor surface drainage in bogs, which rely entirely on precipitation for water inputs [18]. In contrast, fens use groundwater and precipitation water inputs and are generally more resistant to drying [18]. Broadly, the central portion of the study area is underlain by aeolian sand and gravel deposits and has a greater abundance of groundwater-fed fens, while northern and southern extents nearest Great Slake Lake and Lake Claire are a mix of glaciolacustrine clays and basal tills from the retreat of the glacial Lake McConnell [19] that support the formation of bogs (Figure A1). The climate is cold and continental, with a representative climate (1980–2010) with a mean annual temperature of −1.8 °C and precipitation of 353 mm (70% as rain) at Fort Smith (60.00°N, 111.89°W).

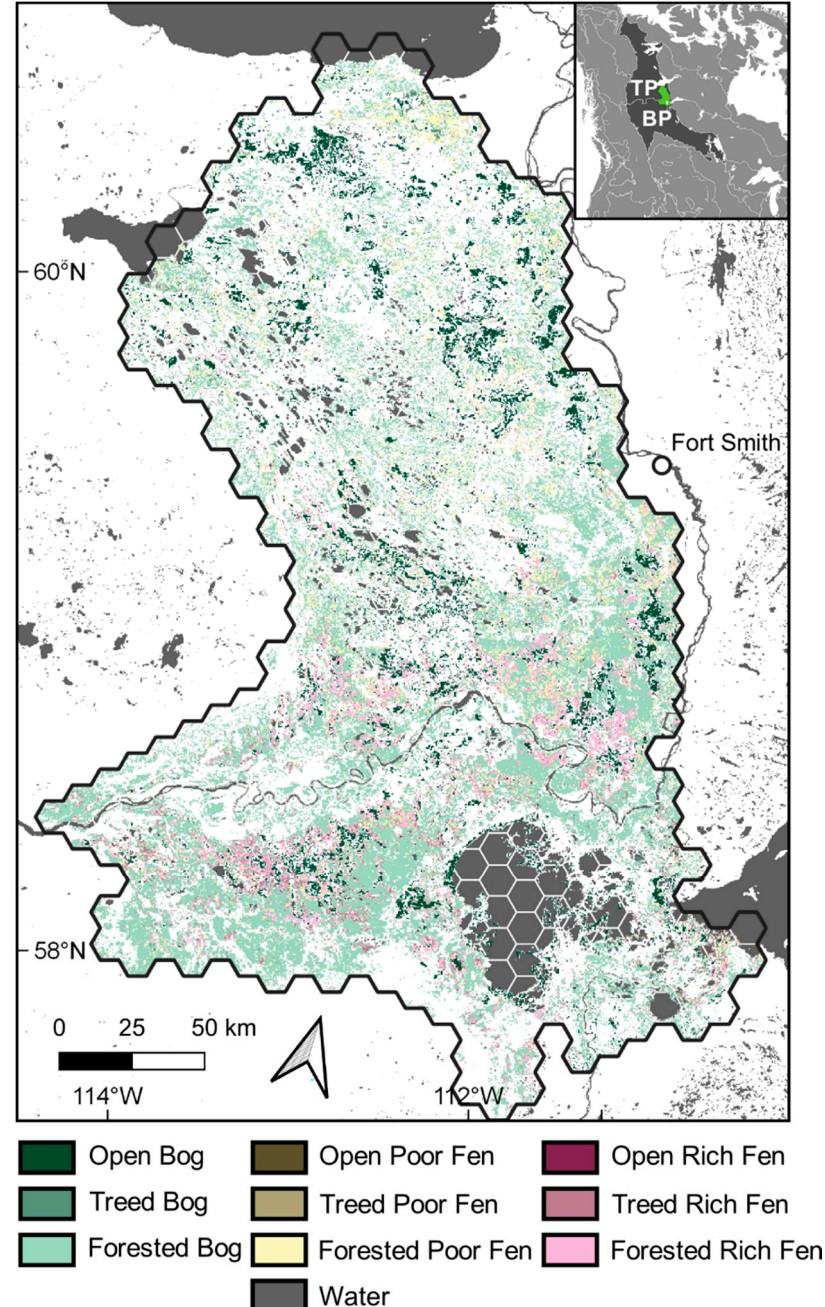

**Figure 1.** Study area and peatland cover. Inset map shows location of study area (green) within Taiga plains (TP) and Boreal plains (BP) North American Level II ecoregions [20]. The 10 kha hexagon grid is shown over water only.

The vegetation of the area is largely composed of upland forests, covering 53% of the region; jack pine (*Pinus banksiana* Lamb.) and trembling aspen (*Populus tremuloides* Michx.) are dominant in more recently burned areas, with a transition to white spruce (*Picea glauca* (Moench) Voss) in older stands. Peatlands encompass 37% of land cover, with forested bogs (black spruce [*Picea mariana* (Mill.) B.S.P.]-dominated) the most prevalent peatland type. Marshes, open-water wetlands, and lakes collectively are mapped as permanent water bodies and make up 9.9% of the study area. A total of 3527 fires have been recorded in the area since from 1980 through to 2017. Lightning ignitions have caused 66% of wildfires in the area. Due to the unequal distribution of fire size, 82 fires (from 6600 to 409,000 ha) have contributed 90% of the 3.7 Mha burned since 1980, of which 96% were ignited by lightning. Of those 82 fires, 70% were ignited in the months of June and July, though many burned

through the summer and received limited suppression only around highly valued resources and assets. Eight years are responsible for 83% (3.1 Mha) of the historical area burned, including the notable multi-year droughts of 1980–1981 (1.3 Mha) and 2014–2015 (0.6 Mha).

*2.2. Mapping Flammable Vegetation*

Fire spread in Canada is modelled using the Canadian Fire Behavior Prediction (FBP) System [21]. The FBP Sysrem links weather and modelled fuel moisture (as described by the Fire Weather Index System) to predict the fire rate of spread in one of 17 fuel types, which are fall within one of the following classes: grasslands (O), upland conifer forests (C), upland deciduous forests (D), conifer-deciduous forest mix (M), or logging residue (S). Since the variety of boreal peatlands are not represented by 17 FBP System fuel types, here we devised a scheme to associate differing peatlands in both wet and dry conditions to the FBP System fuel types that approximate their fire spread rates.

From a wildfire fuels perspective, the groundwater status of peatlands is differentiated between bogs dominated by sphagnum mosses that readily retain water even at low water tables [22], compared to poor and rich fens, which feature deciduous sedge (largely *Carex spp.*) and shrub surface fuels that are sensitive to varying water-table depth. Fens (peatlands with significant groundwater contributions) commonly have a water table closer to the surface than bogs given the same weather conditions [18], but have a surface fuel bed that dries readily when the water table is far from the surface [6]. Moreover, the tree canopy of bogs is almost exclusively composed of more flammable black spruce [23], as compared to a mix of spruce and less flammable eastern larch (*Larix laricina* (Du Roi) K.Koch) in poor and rich fens [6]. To reduce the probability of densely forested conifer uplands being confused with forested bogs in particular, we removed any pixel with pine as a dominant species from the wetland map, as pine-dominated forest in the region is found almost exclusively in uplands [24].

Therefore, a forested bog with abundant surface water may allow for only slow fire spread that can be approximated by allocating forested bog areas to the leaf-on deciduous upland forest (D-2) fuel type. Similarly, a treeless open rich fen may burn at rates approaching a grassland (O-1b) fuel type only during periods of severe drought and water table decline. With no widespread quantitative record of fire spread through differing peatlands across a hydrological gradient, we used a blend of expert opinions of the authors alongside local fire managers to determine FBP System fuel type associations between peatland types and their hydrological status (Table 1). Logically, when an ecosystem has a water table above the surface and surface fuels are saturated by open water, only then can the system be treated as a non-fuel.

**Table 1.** Fuel type attribution by drought scenario and wetland type. Critical Drought Code for the modelled water table being 25 cm below the surface (open and treed rich fens with sedge cover) or 40 cm below the surface (moss and shrub surface cover, all other types) is shown in the bottom row (see Table A1). Mixedwood (M) fuel types note the conifer content (i.e. C50 as 50% conifer). See Table 2 for description of fuel types and fire spread rates. NF stands for "non-fuel" and indicates a peatland ecosystem in a hydrological state with extensive standing water. All other fuel types will allow at least minimal fire spread rates.

| Scenario | Bog Forested | Poor Fen Treed | Bog Treed | Rich Fen Treed | Poor Fen Forested | Poor Fen Open | Bog Open | Rich Fen Forested | Rich Fen Open |
|---|---|---|---|---|---|---|---|---|---|
| Low | D-2 | D-2 | D-2 | NF | D-2 | NF | D-2 | D-2 | NF |
| Moderate | C-2 | O-1a | M-1/2 C50 | O-1a | M-1/2 C80 | NF | D-2 | D-2 | NF |
| Severe | C-2 | O-1b | C-2 | O-1b | M-1/2 C80 | O-1b | D-1 | M-1/2 C65 | O-1b |
| % of study area | 22.5 | 0.1 | 0.8 | 1.1 | 4.1 | 0.0 | 5.0 | 3.6 | 0.0 |
| Critical DC | 250 | 290 | 330 | 400 | 560 | 600 | 640 | 670 | 710 |

The Fire Weather Index System models the moisture content of dead fuels and organic soils of the forest floor via daily noon measurements of surface (2 m) temperature, humidity, wind speed (10 m), and rainfall. Of the three FWI System fuel moisture models (Fine Fuel Moisture Code, Duff Moisture Code, and Drought Code) that drive fire spread in the FBP System, the Drought Code (DC) models the deepest layer, a thick organic soil duff layer in a closed conifer forest, where initial near-saturated soil conditions (DC ~ 0) dry rapidly and exponentially, losing 66% of the initial moisture content after 52 days of constant drying. In the Drought Code moisture model, daily noon temperature serves as a proxy for total soil evaporative demand, and only daily precipitation events greater than 2.8 mm contribute to replenishing soil water at this depth. This Drought Code moisture model is not explicitly designed for modelling the hydrology of a boreal peatland, though it has been found to adequately predict the water table depth of peatlands [16].

Since maps of FBP fuel types alone do not account for differing peatland types, a distinct peatland map of the same resolution must be overlain with a fuels map in order to simulate fire spread. We use a peatland map (Figure 1) [25] that builds upon the forested and treed peatland map derived from the Canadian National Forest Inventory [26] by combining a 250-m forest structure map with ground surface slope and climate variables in order to map peatlands by both groundwater input (bogs, poor fens, and rich fens) and canopy closure (open, treed, and forested).

## 2.3. Dynamic Wetland Drying

In order to estimate wetland drying across a variety of drought levels and peatland types, we developed a simple water table model for boreal peatlands based on DC. We capitalize on the retrieval of archival water table data from a wetland database [27] that features one-time manual water table measurements from 296 sites visited between 1981 and 1989. The original publication of the database in [27] did not feature the exact day of year of sampling; archival retrieval of the original database allowed for the addition of sampling day alongside wetland type, fire weather values, and water table measurements. Sites were classified according to a 9-class framework that accounts for variable canopy closure in open (<10% canopy closure; $n = 124$), treed (10%–25%; $n = 94$), and forested (>25%; $n = 78$) peatlands along a gradient of bog ($n = 84$), poor fen ($n = 44$), and rich fen ($n = 168$).

Non-permafrost peatlands with a recorded water table, primarily from the Boreal plains [28] were used in the fitting of a linear ANCOVA model that accounts for variable slope and intercept according to the nine wetland classes used. The generic structure of the water table decline function follows the form:

$$WT = a_{n,c} \, DC + z_{n,c} \tag{1}$$

where *WT* is the elevation of the water table relative to the peat surface in cm, with negative values below the ground surface, *DC* is the Drought Code (unitless) from 0 and upwards, *a* describes the linear model best fit of the rate of water table decline in cm per unit of Drought Code, and $z_{n,c}$ is a water table offset in cm. Both *a* and *z* may vary by peatland nutrient status (*n*) and canopy closure (*c*); see Table A2 for model coefficients.

## 2.4. Hydrologically Dynamic Fuel Type Assignments

The water table depth model can be related to the moisture content and flammability of surface fine fuels, which play a significant role in fire behavior [28]. We simulate water table dynamics in two broad classes of peatlands: those with moss and shrub surface cover (bogs and some poor fens) and those with sedge graminoid surface cover (largely fens). For peatland ecosystems dominated by mosses and shrubs, water transport from the water table to the surface becomes constrained when water tables decline below 40 cm [29,30], thus allowing surface fuels and organic soils to dry increasingly at equilibrium with the atmosphere. While no definitive discrete water-table threshold exists for the flammability of moss-dominated systems in wildfire, we take this approach informed by soil physics [29] to represent the tipping point of increased flammability. For sedge-dominated

systems where the dominant fuel is sedge litter, no such vertical water transport exists; therefore, the threshold for water table is given to be 25 cm below the surface, which corresponds to the average vertical variability in surface elevation [31]. Hence, this 25 cm threshold represents the percentage at which when the water table has declined sufficiently so that there is no longer surface water present and sedges are able to dry at equilibrium with the atmosphere.

We chose three DC scenarios of 250, 500, and 750, which correspond to break points for multiple peatland fuel assignments to represent low, moderate, and severe drought levels (Table 1). For the 45-year Fire Weather Index System climatology of Fort Smith, NWT, a DC of 250 or greater occurs in 28% of years in the month of May, whereas DC values over 500 occur in 39% of years in the month of August (see Figure A2). DC values neared 750 in 2014 and 2015, years that featured extensive fires in the study area ([15]), and this scenario approximates any of these recently observed extreme drought periods.

When the water table is predicted to decline to a state lower than the critical threshold, we modelled a transition from a less-flammable fuel type to a more flammable one, from within the 17 possible fuel types of the existing FBP System (Figure A3) [21]. Fuel type classifications in the drier state beyond the critical DC were guided by the fundamental fuel structure, and influenced by the personal field experience of the authors. Peatlands dominated by boreal mosses do not typically experience complete saturation, and areas remain above the water table, though surface soil moisture content can be upwards of 40% volumetrically. Therefore, these systems were classified as leaf-on deciduous forest (D-2), where fire spread is possible, but the intensity and spread rates are severely limited [32]. As the critical DC threshold is exceeded amongst various peatland classes, the transition is to a fuel type more similar to the vegetation structure with a higher fire spread rate (i.e. boreal conifer C-2 in forested bogs). In sedge-dominated peatlands (open and treed rich fens), the surface vegetation is highly adapted to saturated conditions, where the water table can be upwards of 50 cm above the soil surface. During periods of low DC, we modelled these systems as being almost entirely saturated, and therefore classified them as non-fuels incapable of any fire spread. With declining water tables below the surface, the sedge litter is available to burn similar to a grassland fuel (O-1a). For forested rich and poor fens that are a mix of spruce and larch stands, we used the average larch content over the study area in the respective mapped peatlands (35% and 20% in rich and poor fens), equal to 65 and 80% spruce content, such that the stands are considered a mix of deciduous and conifer forests. The resulting fuel grids corresponding to the three drought scenarios are shown in Figure 2.

*2.5. Burn Probability Simulations*

Burn-P3 is a Monte Carlo simulation model based on the Prometheus fire spread engine [33] that simulates the ignition and spread of fires across the landscape. Burn-P3 combines deterministic fire spread with probabilistic fire ignition locations, fire duration, and weather [34]. We assembled Burn-P3 inputs using methods described in detail in [35]. We calibrated model inputs to provide simulated fire outputs that most closely match fire history information from the period 1961–2016 to ensure present-day realism in model outputs. Each Burn-P3 iteration simulated a single independent wildfire ignition. The primary model output considered in this study is burn rate, defined as the number of times a pixel will experience fire relative to the total number of iterations; and (b) simulated fire perimeters and their ignition points.

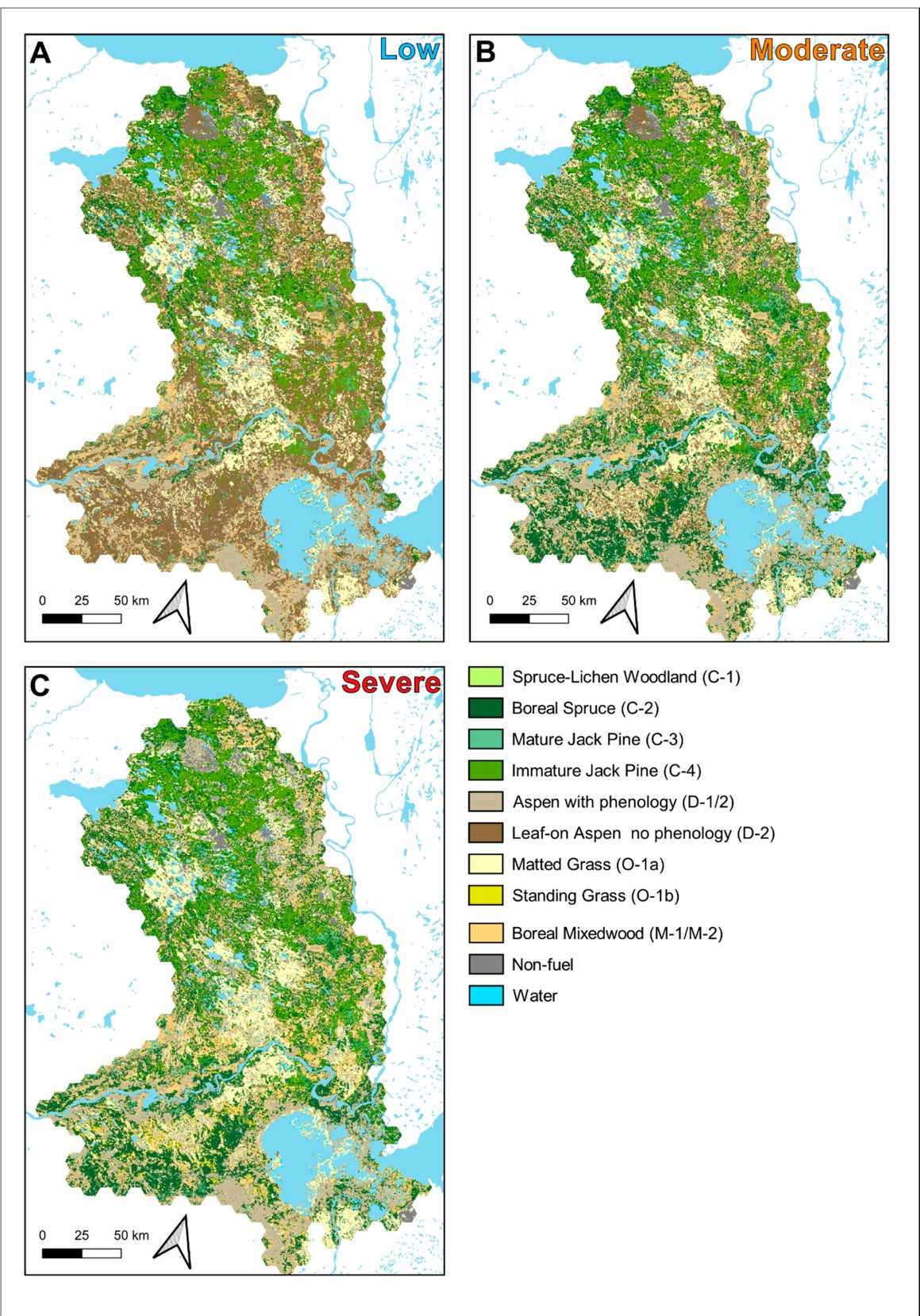

**Figure 2.** Canadian Fire Behavior Prediction System fuels maps corresponding to the differing drought scenarios (**A**) low; (**B**) moderate; (**C**) severe.

The fire regime of the study area can be divided into two distinct periods: spring and summer. In spring, human ignitions dominate and deciduous forests have not yet leafed out and may carry significant fire spread; open grass fuels (including sedge fen peatlands) carry fire through the sedge litter bed. The summer fire season is dominated by lightning ignitions, with only limited fire spread through deciduous forest and little to no spread in grasslands. Two seasons were generalizedas a spring season from 1 May—15 June, and a summer season from 15 June to 31 August. Grass curing in O-1 fuels was set to 80 and 65 percent in the two seasons, respectively. Fire weather records from Fort Smith, NWT from 1980 to 2016 were used in the analysis through the entire model domain. Daily fire weather data from the historical record were used, where the Initial Spread Index (ISI, a proxy for spread rate representing an integration of dry fine fuels, wind, and low relative humidity) was greater than 8.5; this corresponds to days of probable fire spread in the region [35]. Identical daily weather and ISI values were used across all drought scenarios, as the distribution of ISI does not vary during drought (Figure S3). The length of the burning period was determined by sampling from a distribution of the historical number of spread days (days where the fire grows significantly under favorable weather conditions), and fitted to distributions used in previous studies from the region [35]. Fire weather records were drawn from consecutive days from the historical record. Fire weather was drawn randomly from the same source dataset for all three drought scenarios, with no sub-sampling of daily fire weather and associated temperature, humidity, and wind speed values associated with specific drought conditions (Figure S3).

In order to isolate the effect of fuel type in the simulations, ignitions were placed randomly across the landscape, contrary to observed historic patterns [36]. Initially, 100,000 ignitions were simulated using the severe drought (Drought Code 750) scenario. Ignitions that occurred in non-flammable areas (water, salt plains, rock, or areas less than 3 years since last wildfire) were considered immediately extinguished. Any ignitions from the severe drought scenario that did not achieve 10 ha in size due to a dynamic change of peatland fuel status to non-fuel or less flammable fuels were not recorded as a successful ignition. A total of 13,550 ignitions either occurred in non-fuel areas or failed to achieve 10 ha in size, resulting in 86,450 total ignitions greater than 10 ha in size for the severe drought scenario. The Burn-P3 'replay' function was then used to simulate the exact same fire ignition locations and weather under the Drought Code 500 (moderate drought) and 250 (low drought) scenarios where fuels were modified to less flammable conditions in wetlands. As the fraction of non-fuel areas increased due to the increasing water table in wetlands, the number of ignitions was reduced to 78,760 and 56,130 in the moderate and low scenarios, respectively. In order to provide a fine-scale perspective on burn probability within the study region's locally variable wetland abundance, 481 hexels each approximately 10,600 ha in size (roughly 11 km across). The sum of the burned area per hexel was then pro-rated by fuel area (i.e. area not permanent water or permanent non-fuel) in order to derive an averaged burn rate in each hexel per 100,000 ignitions in a given scenario:

$$burnrate = \frac{\sum_{i=1}^{n} n_b}{1 - Aw - A\text{NF}} \tag{2}$$

where for pixels $i$ through $n$ in a given hexel, $n_b$ is the number of burns per pixel, $A_w$ is the fraction of water area in the hexel, and $A_{NF}$ is the fraction of permanent non-fuel (i.e. sand, rock, salt plains). In order to assess fuel control on burn rate, a generalized linear model (GLM) was created [37], using per-hexel burn rate (dependent) and the proportion of differing land covers per hexel (peatland, aspen, non-fuel, and lakes, as independent variables) as computed using a Poisson count variant in R version 3.4. The land cover fraction independent variables were largely not correlated with each other, save a moderate inverse correlation between peatland and water area (Table S2). The model was run for each drought scenario independently. Marginal effects of differing land cover fractions were determined by solving the linear model over a gradient of land cover and drought conditions. To overcome spatial autocorrelation, each GLM was constructed 100 times, each iteration using a random 50% subset of the hexels. Model responses were collected from each iteration, and the 5th, 50th, and 95th percentiles

of each model parameter from across the 100 iterations were calculated in order to compute the 95% confidence band for the model response.

## 3. Results

### 3.1. Water Table Prediction

Without the benefit of supplementary information on peatland canopy cover and nutrient class, Drought Code itself is poorly related to observed water table (Adj. $R^2 = 0.07$; $F_{1,294} = 26.1$). However, in an ANCOVA model including variable intercepts by canopy and nutrient classes, fit is significantly improved (Adj $R^2 = 0.51$; $F_{5,290} = 64.0$, Table S2). The inclusion of a variable slope per nutrient class yields a smaller slope for rich fens and a smaller residual sum of squares compared to a fixed slope model (not shown). Given that the lower slope in fen peatlands makes ecological sense given the extra groundwater inputs, we chose the more complex of the two models for use in the framework. The slope of the water table (cm below surface) in rich fens as a response to a unit increase in Drought Code was $0.041 \pm 0.012$ ($\pm$ 1SE, $t = 3.4$, $p < 0.001$), whereas the slope response of poor fens ($0.043 \pm 0.023$) was not meaningfully different from the bog baseline $0.046 \pm 0.015$. Open sites show a significantly higher water table intercept relative to the bog baseline (+ 16.6, $t = 8.1$, $p < 0.001$), whereas treed sites show no significant difference in water table intercept relative to forested bogs (+ 3.2, $t = 1.6$, $p = 0.10$). Ecologically, this implies that all peatland sites had essentially the same rate of water table decline per unit increase in Drought Code. Open (treeless) peatlands showed a consistently high water table across all bogs and fens; the treeless status of these peatlands is likely owing to their hydrological state of more frequent surface saturation. Fens (both rich and poor) have a significantly higher water table given the same Drought Code value on the landscape, again an indicator of increased groundwater inputs in fens [38] and the requirement for drier conditions in fens before critically dry conditions for fire spread are reached.

### 3.2. Fuel Distributions under Varying Drought Conditions

Bogs, largely forested, form 74% of all wetlands (28% of the study area), followed by rich fens, which make up 15.6% of all wetlands in the study area. The moderate drought scenario converted 17% of the landscape from leafed out deciduous D-2 (creeping surface fire only) to C-2 boreal spruce as forested bogs became dry enough to carry significant wildfire spread (Table 2, Figure S1). In the severe drought scenario, C-2 area increased by only 1%, but less flammable fuels including D-1 leafless aspen (5%), O-1b standing grass (1%), mixedwood with 65% conifer (4%) were added to the landscape, reducing the total area of D-2 and non-fuel from 21% to 11% in the DC500 fuel landscape.

### 3.3. Wetland Fuels Iimpacts on Burn Probability

The burn rate (sum of burned area per hexel normalized to exclude permanent non-fuels such as water or rock) per 10 kha hexel varied from 0 to 150 in the low scenario, with a median of 21.4 and interquartile range between 6 and 60. Accordingly, as wetland fuels are converted from non-fuel or low-intensity fuels to those capable of supporting significant fire spread, the median burn rate across all hexels increased by 530% and 790% in the moderate and severe scenarios, respectively, compared to the low drought baseline (Figure 3). The severe scenario saw only a modest proportional increase in burn rate, with many areas in the central and northern portion of the study area showing little increase in simulated fire activity (Figure 3).

**Table 2.** Fractional cover of fuel type per drought scenario (Low, Moderate, Severe), including both wetland (dynamic to drought) and upland (static) fuels. For instance, mixedwood 50% conifer includes both upland, static areas (11.4%), and an additional 0.8% of the landscape converted to M–2 with 50% conifer only in the moderate scenario.

| Fuel Type | Low | Moderate | Severe | Rate of Spread * |
|---|---|---|---|---|
| C-1 spruce-lichen woodland | 0.2 | 0.2 | 0.2 | 2 |
| C-2 boreal spruce | 8.1 | 25.2 | 25.9 | 10 |
| C-3 mature jack pine | 2.0 | 2.0 | 2.0 | 3 |
| C-4 immature jack pine | 12.3 | 12.3 | 12.3 | 11 |
| D-1 leafless aspen | 0.0 | 0.0 | 5.0 | 2 |
| D-2 green aspen | 30.7 | 8.7 | 0.0 | 0.4 |
| D-1/2 aspen with phenology ** | 8.0 | 8.0 | 8.0 | 0.4 |
| O-1a matted grass | 13.0 | 14.1 | 13.0 | 3 |
| O-1b standing grass | 0.0 | 0.0 | 1.2 | 7 |
| Non-fuel | 4.4 | 3.3 | 3.3 | 0 |
| Water | 9.9 | 9.9 | 9.9 | 0 |
| Mixedwood 50% conifer | 11.4 | 12.2 | 11.4 | 5.5 |
| Mixedwood 65% conifer | 0.0 | 0.0 | 3.6 | 7 |
| Mixedwood 80% conifer | 0.0 | 4.1 | 4.1 | 8.5 |
| Total open (O) | 13.0 | 14.1 | 14.2 | - |
| Total conifer (C) | 22.6 | 39.6 | 40.4 | - |
| Total mixedwood (M) | 11.4 | 16.2 | 19.2 | - |
| Total deciduous (D) | 38.7 | 16.7 | 13.1 | - |
| Total D-2 + non-fuel | 35.1 | 12.0 | 3.3 | - |

* Rate of spread (m min$^{-1}$) for a minimum spread day (Initial Spread Index 8, Buildup Index 70). Spread rates for D and M fuel types represent summer (leaf-on) conditions and 65% curing in O fuel types. ** Aspen with phenology refers to a fuel type that is leafless (D-1) in spring and transitions to leaf-on (D-2) in summer. This is in contrast with the above fuel types that remain as D-1 or D-2 throughout the entire year.

Median simulated fire size (as measured in a random subset of 1000 fires) in the low drought scenario was 137 ha, with a 95th percentile fire size in the low scenario of 11,400 ha. Median fire size in the moderate drought scenario increased to 725 ha, significantly larger than the low scenario (Wilcoxian rank sum test, $n = 1000$, $W = 289{,}400$, $p < 2.2 \times 10^{-16}$). The 95th percentile of the fire size in the moderate scenario changed at a similar rate to 58,000 ha. Simulated median fire size in the severe scenario was modestly, but significantly larger than that of the moderate scenario at 900 ha ($n = 1000$, $W = 532{,}750$, $p = 0.011$) and 70,000 ha, respectively.

The linear model of burn rate per hexel as explained by differing land cover fractions captured 65%, 50%, and 63% of the variance in the landscape simulations data for the low, moderate, and severe drought scenarios, respectively. The marginal effects of peatland and aspen extent computed from the model (Figure 4) allow for the comparison of burn rates between hexels of differing composition against the GLM intercept that represents a continuous upland forest stand with no water, non-fuels, peatlands, or aspen. The effect of permanent water bodies in the GLM was not constant across all drought scenarios, where an idealized hexel with 50% water resulted in an 82%, 67%, and 62% reduction in burn rate across the low, moderate, and severe drought scenarios, respectively. Permanent non-fuels (such as rock and recent burned area) showed a relatively constant reduction in burn rate across all scenarios, where an idealized hexel with 50% permanent non-fuel (typical of areas of patchy recent burned areas) was simulated to have a 70–74% reduction in burn rate regardless of drought scenario (Figure 4). In the low drought scenario, the abundance of aspen in some regions (Figure A4) outweighed the impact of peat in determining the burn rate, where a hexel containing half aspen is predicted to have a burn rate only 6% compared to the model intercept (an entirely upland forest), whereas a half-peatland hexel will burn at 14% of the rate of the model intercept.

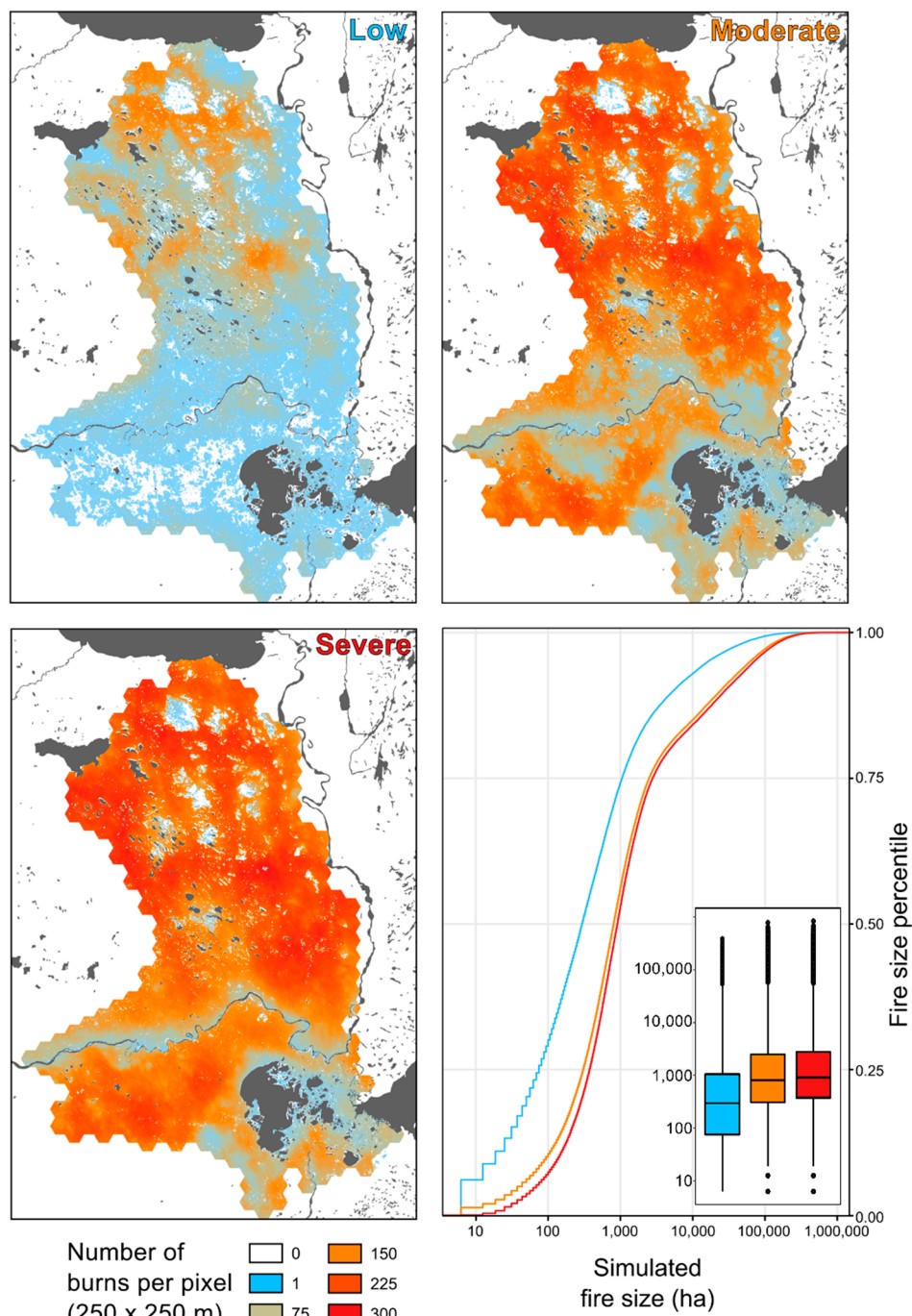

**Figure 3.** Burn rate (number of burns per 100,000 ignitions) across the study domain as a function of drought scenario: low 250), moderate (DC 500), or severe (DC 750). Cumulative fire size distributions for the respective scenarios are also shown on a $\log_{10}$ scale with inset boxplots of the same data.

Permanent water and non-fuels had only slightly weaker impacts on burn rate per unit area compared to peatlands (an 85% reduction in burn rate for a hexel half covered by water or other non-fuel), but are on the whole far less abundant than peatlands or aspen on the landscape (Table 2). In the moderate drought scenario (Figure 4), peatland area per hexel showed far less impact on the burn rate than under low drought conditions, with a hexel of 50% peatland reducing the burn rate by 28% compared relative to a hexel with zero peat experiencing the same weather. Aspen abundance also showed a reduced impact on burn rate, increasing from a rate of 6% of the magnitude of the intercept (low drought) to 36% (moderate drought). In the severe drought scenario, hexels consisting of half

peatland showed only a 10% reduction in burn rate compared with an upland conifer stand, whereas aspen impacts on burn rate remained nearly identical to moderate drought conditions.

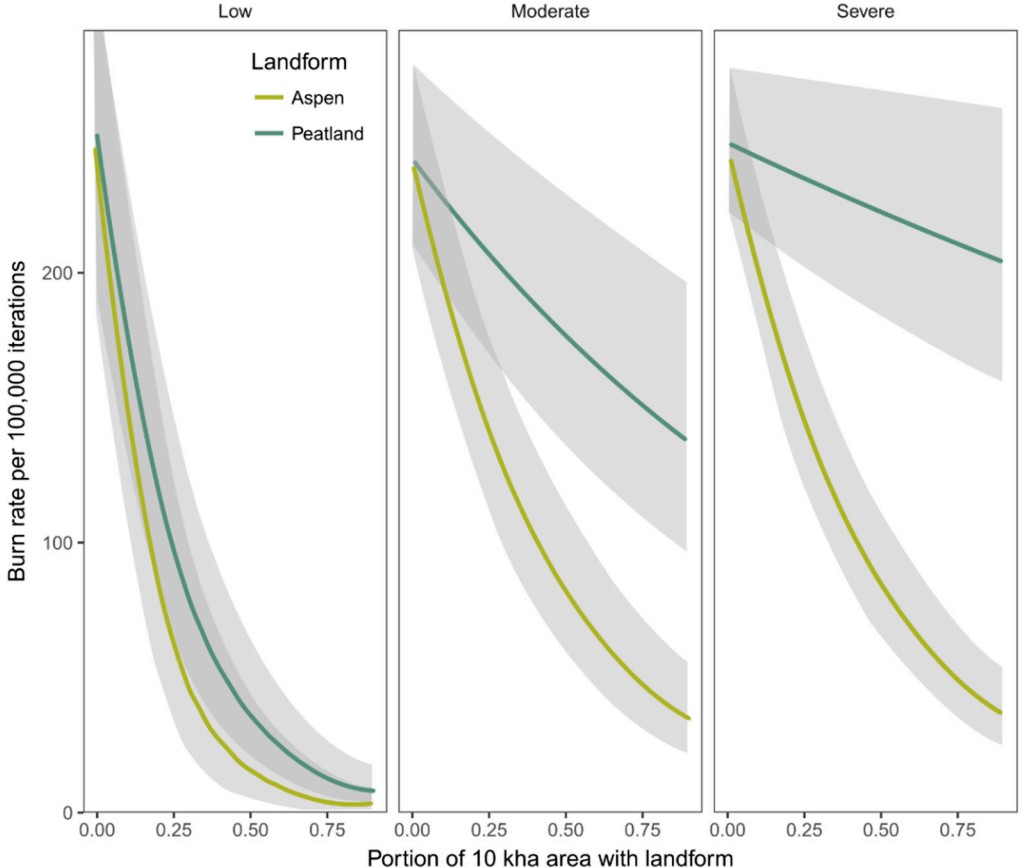

**Figure 4.** Generalized linear model of burn count per 10 kha hexel, showing the effect of aspen and peatland fractional cover on the burn rate in each drought scenario. The model intercept, at zero peatland and aspen cover, represents the burn rate in a hexel composed entirely of conifer, mixedwood, and non-peatland grasslands corresponding to average landscape values (Table 2). The 95% confidence interval of the prediction is shown for each model response curve.

## 4. Discussion

The frequency and size of large fires in Western North America has increased as a result of anthropogenic climate change [39,40]. Further increases in fire activity are projected globally, as warming and drying of climates increase vegetation flammability, alongside increases in lightning frequency and human ignitions [12,41,42]. The results of this study show how, when their characteristic surface saturation is lost through prolonged droughts, biomass-rich wetlands (i.e., peatlands) are no longer an impediment to wildland fire spread. Specifically, once a certain drought threshold is overcome, naturally fragmented landscapes of the boreal plains become functionally homogeneous—from a fire behavior standpoint—and experience some of the consistently largest wildfires in North America. Although peatlands (or other wetland types) of different biomes may exhibit different dynamics than those described in this study, it may be misleading to consider these systems as non-flammable. A telling is example is that of the 1997 extensive peatland fires in Indonesia, which caused health concerns in much of Southeast Asia and released massive amounts of carbon in the atmosphere [43].

Peatlands cover nearly 3% of terrestrial landscapes [1] and collectively store a reservoir of carbon roughly equal to that in the atmosphere [44]. As fire activity and drought increase concurrently, it is of global importance to understand the effect of wetlands on fire activity. Even in the absence of fire, North American peatlands under climate change have been shown via paleoecological analyses to

be shifting away from less-flammable open and treed fens and towards enhanced *Sphagnum* moss cover [45] that may ultimately lead to a greater abundance of more-flammable treed and forested bogs. While the ground vegetation community of boreal peatlands is largely resilient to wildfire disturbance [46], frequent, high–severity fire decreases the abundance of the deciduous *Larix laricina* in treed and forested rich fens [47], which serves to dampen fire spread through high foliar moisture content [6]. Landscape heterogeneity has the potential to dramatically alter fire activity due to the contagious nature of fire as a process [48]. In this research we have quantified the dampening effect of wetlands on fire activity on a real landscape, demonstrating that when surface water tables are lowered, thereby removing the effective barrier to fire spread and ignition from wetlands, fire activity (burn rate) increased by at least 50% and fire sizes increased by a factor of five, with all other factors (i.e., weather and ignitions) remaining equal. Wetland effects on fire activity are not limited to these areas, but also project onto adjacent uplands as emergent properties of the system. From a fire management standpoint, considering wetlands as static barriers to fire spread and failing to account for altered hydrological conditions may lead to gross misinterpretations of landscape fire hazard.

Aspen and other deciduous forest is often considered alongside peatlands as a part of the boreal forest landscape less capable of carrying fire [49]. Since aspen and deciduous forests generally havefar less surface organic soil available for combustion [21,32] and are generally disconnected from groundwater systems sensitive to drought [18], no such drought-sensitive fuel type changes were simulated in upland forests. Even with this static representation of aspen forest flammability in the Burn-P3 model, a marked increase in the burn rate is observed in the moderate drought scenario for aspen-dominated forests (Figure 4). This increase in non-peatland burn rate suggests a landscape contagion where increasing burning of peatlands impinges on adjacent aspen forests, especially where the two ecosystems intersect more frequently in the southern half of the study region (Figure A5) and where the largest increases in burn rate are observed (Figure 3).

This work provides insights on how a landscape naturally fragmented by wetlands can occasionally reconnect and allow for the spread of some of the very large wildfires that have been observed during the modern era in northwestern Canada. Evidence of wetland flammability as a contributor to large fire events has been either presented (in detail) for a single fire [50] or, conversely, was carried out at a coarse spatio-temporal time scale [49]. The phenomenon of drought contributions to very large fires was displayed in the exceptional 2014 fire year in the Northwest Territories, where 3.4 Mha burned [15]. It is these landscape dynamics including the drying of peatlands [38] that contributed to the 590,000-ha wildfire that burned much of the town of Fort McMurray, Alberta, which became the costliest natural disaster in modern Canadian history. The processes described here highlight the nonlinear effects of drought on landscape fire activity in the boreal biome, pointing to what is likely a key threshold under which the boreal landscape reconnects from a fire spread perspective. Our current approach is limited to three drought scenarios each with 100,000 fire simulations; future work may involve a more continuous treatment of landscape drought in order to determine key drying thresholds at large spatial scales. The phenomenon presented in this study has been surmised for some time [51,52]; here we put forward the specific mechanism that can inform short-term and seasonal forecasts by fire management agencies. Other than a peatland map, this framework required no further input of new data layers, and rather relies on an algorithmic interpretation of fuels and fire weather in such a way that can be incorporated relatively easily into operational fire management systems. Similar schemes to enhance fire danger assessment in Canada include refinements to moisture-retention models (the Sheltered Duff Moisture Code; [13]) and various lightning and human fire occurrence prediction systems ([53,54]), where the focus lies on enhancing existing fire danger rating systems while adding a minimum of new calculations and spatial data.

A more realistic representation of fire likelihood during drought will lead to an enhanced understanding of wildfire risk to highly valued resources in these wetland-abundant areas. The dramatic differences in both the overall magnitude and spatial patterns of the burn probability maps under drought conditions provide a compelling argument for the inadequacy of static fuels maps for assessing

wildfire potential for operational (i.e., day-to-day) purposes. The highly dynamic nature of the boreal plains landscape, recognized for decades in the field of hydrology [10], in conjunction with the nonlinear dynamics of disturbance regimes [55], preclude suitable compromise for a static representation of these peatland fuels. Wildfire managers have an increasing need to be aware of when active fires under moderate conditions are unlikely to threaten values, allowing for the use of fewer resources and achieving more burned area under moderate fire weather conditions and severity. For instance, the Horse River fire of 2016 impacting the community of Fort McMurray, Alberta, showed extreme fire spread in upland fuels, but the extent of burning in peatlands was more limited due to the moderate drought conditions (DC of 488) that limited rapid spread to bogs, and not the extensive poor and rich fens of the region [38].

Although this study successfully demonstrates how including real-world fluctuations in landscape connectivity under different drought levels affects fire-spread potential, it remains a simplification of a phenomenon of much greater complexity. By design, other biophysical factors that may influence wildfire were held constant in order to isolate the effect of peatland drying, but the state of non-wetland vegetation may also exhibit complex and nonlinear relationships with weather. For instance, whereas young forest stands (<50 years old) tend to resist fire spread compared to older ones, this resistance will be overcome during particularly severe weather conditions [56,57]. Similarly, by focusing on spread potential, the design of the study neglects the effect of drying fuels on ignition potential, which is highly sensitive to drought, though this varies greatly among vegetation types [28]. In other words, the effects of drought on fire probability reported in this study is almost certainly underestimated by downplaying the role of ignitions.

Climate change has the potential to dramatically increase the area burned in the boreal forest, as the frequency and severity of drought is likely to increase [11,58]. In the study area, such increases will build from an already-frequent baseline (the "moderate" drought scenario we simulated recurs as regularly as every five years). Although drought did not relate to a substantial increase in severity of other fire weather drivers (i.e., wind speed), the increasing connectivity of the landscape under such conditions may substantially increase fire activity and size, as modelled here. Just as drought conditions will increasingly reduce the extent and persistence of barriers to fire spread, increasing drought frequency under climate change will enable smoldering combustion in peatlands, releasing legacy carbon and accelerating climate change [59]. In this context, increasing fire size associated with lowered wetland water tables may have globally important implications for forest ecology and fire management.

## 5. Conclusions

The results of this study capture some of the spatio–temporal interactions among wetlands, upland vegetation, and nonfuel features of the landscape. Whereas our results demonstrate how change in flammability of wetlands will invariably affect burning in upland fuels, interactions with nonfuel features are more difficult to quantify. Wildfire indeed exhibits complex relationships as a function of patterns of unburnable features in the boreal forest, as characterized by the interplay of the proportion of nonfuel, its orientation relative to incoming wildfires, and relative isolation (e.g., islands in large lakes). These relationships are greatly complicated when wetland dynamics and fire-weather severity are incorporated. Although much importance has—deservedly so—been given to weather forecasting, given the vast extent of some wildfires of the boreal plain (often > 100,000 ha), identifying specific landscape contexts conducive to large wildfires is critical in providing effective early-warning systems for land managers. By combining empirical results (water-table measurements), a well-understood fire behavior system (the Canadian FBP System), and wildfire simulation modeling, the results from this study provide a framework for integrating the synergistic effects of peatland drought on landscape flammability.

**Supplementary Materials:** The following are available online at http://www.mdpi.com/1999-4907/10/7/534/s1.

**Author Contributions:** Conceptualization: D.K.T., B.N.S., and M.-A.P.; Data curation, Q.E.B.; Formal analysis, D.K.T., E.W., and Q.E.B.; Investigation, D.K.T. and E.W.; Methodology, D.K.T., B.N.S., E.W., Q.E.B., and M.-A.P.; Resources, M.-A.P.; Writing—original draft, D.K.T., B.N.S. and M.-A.P.; Writing—review and editing, D.K.T., B.N.S., E.W., Q.E.B., and M.-A.P.

**Funding:** This research received no external funding.

**Acknowledgments:** We thank Arlene Hilger for digitizing old water table records; Ilka Bauer assisted in classification of historical peatland sites for the water table model. Jean Morin and numerous other Parks Canada staff assisted in past studies and numerous flights over the area that assisted with the conceptual model and wetland maps in this study.

**Conflicts of Interest:** The authors declare no conflict of interest.

**Appendix A**

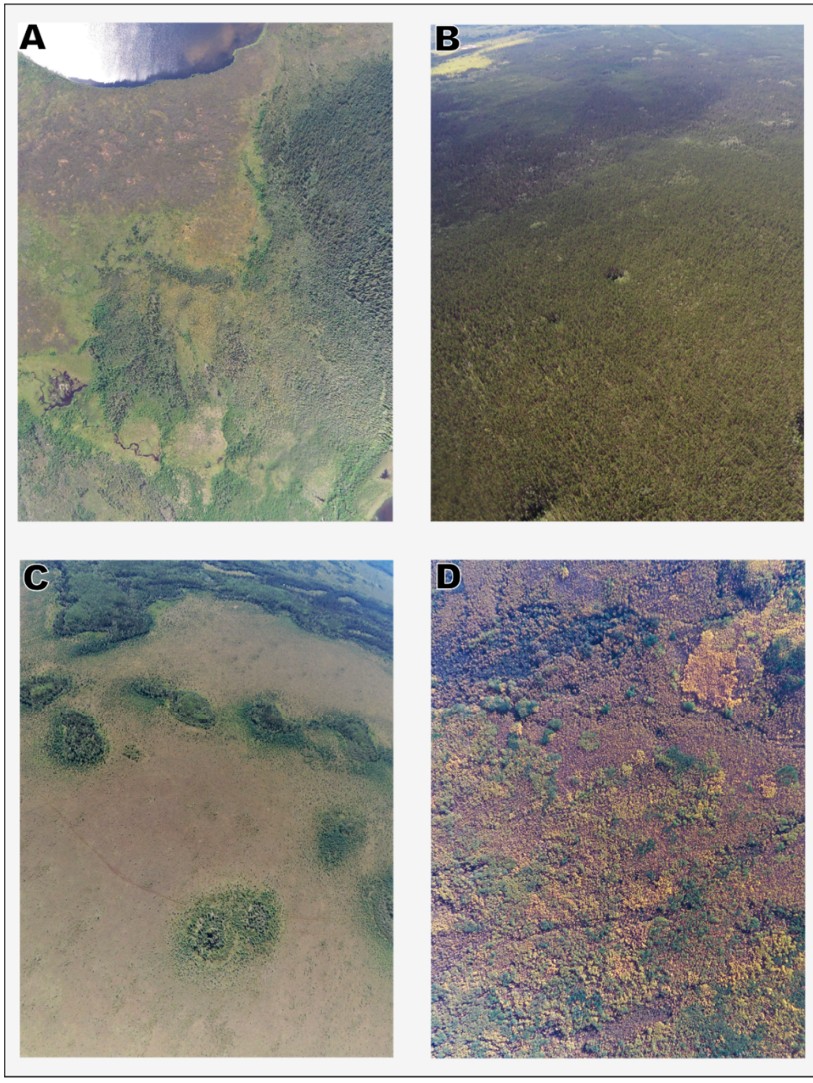

**Figure A1.** Aerial pictures of study area and typical landscapes across differing ecodistricts (see Figure 1). Panel (**A**) small lakes mixed with open bogs, treed bogs, and upland conifer in the Buffalo River Plains; (**B**) continuous pine and mixed conifer in northern Knight Creek Plain ecodistrict; (**C**) extensive open rich fens mixed with upland islands in the Fox Lake Plain; (**D**) continuous deciduous forest on well-drained slopes with low peatland abundance (Figure 4) in the Birch Hills Fans. Photos in panels (**A**) and (**C**) were taken 300 m above ground level; (**B**) and (**D**) were taken 700 m above ground level.

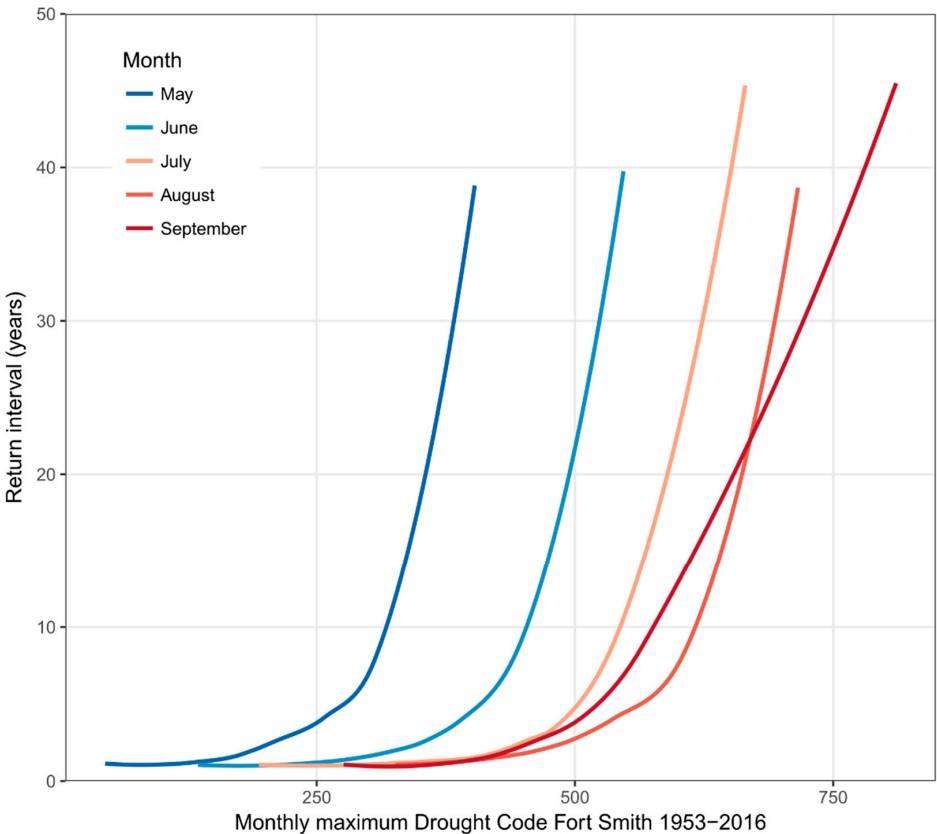

**Figure A2.** Historical distribution of maximum monthly Drought Code in the study area at Fort Smith, with unique distributions for each calendar month of the fire season. Annual return interval was calculated from the historical method using the Gumbel (1941) variant of the Generalized Extreme Value distribution and smoothed using a loess function for visual clarity.

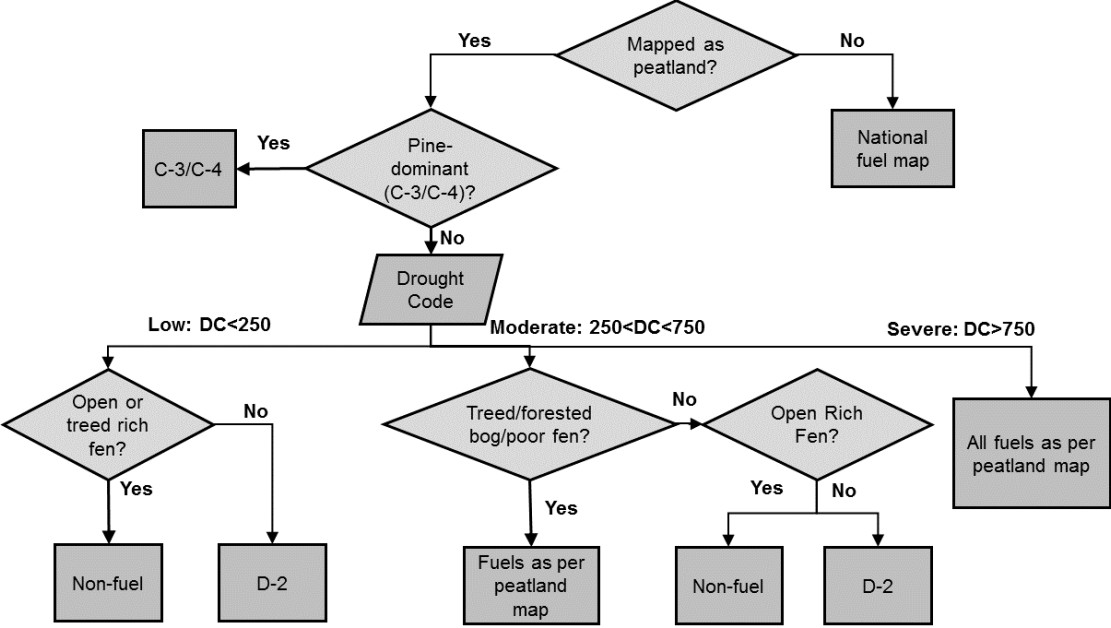

**Figure A3.** Flow chart summarizing the fuel mapping process based on upland fuels maps and the peatland type map and drought scenario. See Table 1 for complete description of the algorithm broken down by all peatland types and Drought Code levels. Table 2 for a comparison of potential fire behavior between fuel types and text description of fuel type (i.e. C–3 mature Jack or Lodgepole pine).

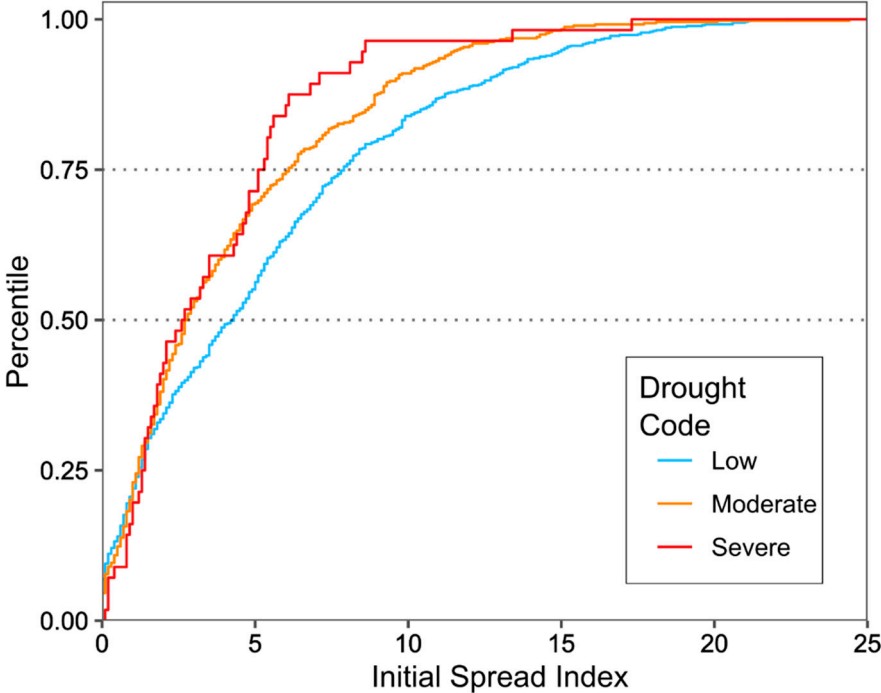

**Figure A4.** Historical cumulative distribution of the daily fire spread potential, Initial Spread Index, at Fort Smith from 1970 to 2015, stratified by Drought Code scenario.

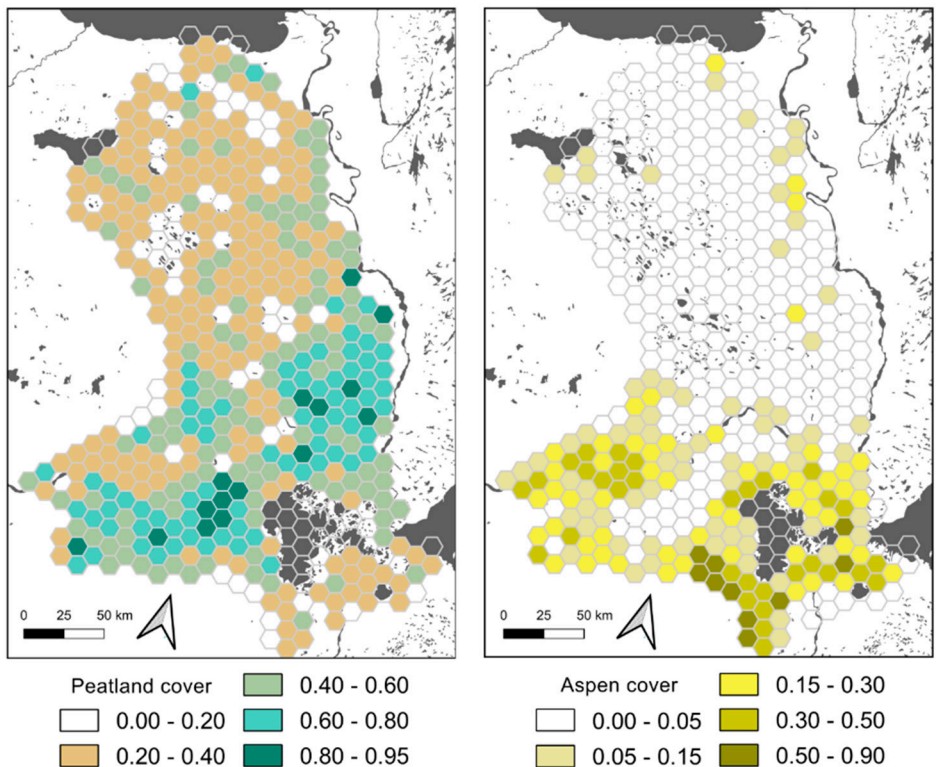

**Figure A5.** Fractional cover of peatland and aspen per 10 kha hexel. Breaks for peatland and aspen land cover classes were calculated using the Jenks method. Peatland cover <20% and aspen cover <5% is shown as a transparent hexel.

**Table A1.** Nominal properties of peatland types used in the peatland map. Water table values are estimates from the ANCOVA model (Table A2) and are relative to the average surface level. Negative values indicate water table below the ground surface.

| Type | Tree Cover | *Larix laricina*% | Water Table (cm) | | |
|---|---|---|---|---|---|
| | | | DC 250 | DC 500 | DC 750 |
| Open bog | < 10% | <25% | -23 | -34 | -46 |
| Treed bog | 10–25% | <25% | −37 | −48 | −59 |
| Forested bog | >25% | <25% | −40 | −52 | −63 |
| Open poor fen | <10% | 25–50% | −10 | −21 | −32 |
| Treed poor fen | 10–25% | 25–50% | −24 | −35 | −46 |
| Forested poor fen | >25% | 25–50% | −27 | −38 | −49 |
| Open rich fen | <10% | >50% | −5 | −16 | −26 |
| Treed rich fen | 10–25% | >50% | −19 | −29 | −39 |
| Forested poor fen | >25% | >50% | −22 | −32 | −43 |

**Table A2.** ANCOVA general linear model for predicting water table depth (cm below surface) from peatland canopy cover, nutrient class, and the interaction between nutrient class and Drought Code. Note that for this GLM, the intercept-only case is a forested bog, so no coefficients are given for bogs or forested peatlands (integrated into intercept). This model captures 52% of the variance in the dataset.

| Coefficient | Estimate | Std Error | *t* Value | *p* |
|---|---|---|---|---|
| Intercept | −28.4 | 4.4 | −6.4 | <0.001 |
| Open | 16.6 | 2.0 | 8.1 | <0.001 |
| Treed | 3.2 | 2 | 1.6 | 0.10 |
| Poor fen | 12.5 | 7.2 | 1.8 | 0.086 |
| Rich fen | 16.8 | 5.4 | 3.1 | 0.0019 |
| Bog:DC | −0.046 | 0.015 | −2.9 | 0.003 |
| Poor fen:DC | −0.044 | 0.023 | −1.8 | 0.064 |
| Rich fen:DC | −0.041 | 0.012 | −3.4 | <0.001 |

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
