# Peer review of "Peatland Hydrological Dynamics as A Driver of Landscape Connectivity and Fire Activity in the Boreal Plain of Canada"

_forests, doi:10.3390/f10070534_

Round 1

Reviewer 1 Report

This is interesting paper presenting quite interesting prediction of fires impact on the wetlands ecosystems in the boreal zone of North America. The aims are clear and have been achieved.

I feel it is worth to publish in Forests after including some paleo data.

The authors predict the spread of fires and the disappearance of barriers to its spreading as a result of long-term droughts. One of the consequences of the decrease of water level are plant population shifts in wetland ecosystems e.g., the spread of the Sphagnum mosses, which was observed in the Boreal (Loisel, J., Yu, Z., 2013) and Arctic Alaska (Galka et al., 2018).
In my opinion, the authors of this study should include the results and observation from palaeoecological data for introduction and discussion chapter. 

The resistance to fire and regeneration after fire of plant populations on peatlands would be particularly important. It is common knowledge that fire is a threat but also an opportunity for some ecosystems.

The authors may take a deeper look at papers e.g. Kuhry 1994; Magnan et al., 2012; Camil et al., 2009 and papers published by Lukenbach et al., 2015, Journal of Hydrology 530: 405-418 and J Hydrology 548: 741-753.

References

Loisel, J., Yu, Z., 2013. Recent acceleration of carbon accumulation in a Boreal peatland,
south central Alaska. J. Geophys. Res. 118, 41–53.

Galka, M., Swindles, G. T., Szal, M., Fulweber, R., Feurdean, A., 2018. Response of plant communities to climate change during the late Holocene: Paleoecological insights from peatlands in the Alaskan Arctic. Ecological Indicators, 85, 525–536

Kuhry P., 1994. The role of fire in the development of Sphagnum dominated peatlands in western boreal Canada. Journal of Ecology 82: 899–910.

Magnan GM, Lavoie M and Payette S., 2012. Impact of fire on long-term vegetation dynamics of ombrotrophic peatlands in northwestern Québec, Canada. Quaternary Research 77: 110–121.

Camill, P., Barry, A., Williams, E., Andreassi, C., Limmer, J., Solick, D.: Climate-vegetation-fire interactions and their impacton  long-term  carbon  dynamics  in  a  boreal  peatland  landscapein northern Manitoba, Canada, J. Geophys. Res., 114, G040.

Author Response

This is interesting paper presenting quite interesting prediction of fires impact on the wetlands ecosystems in the boreal zone of North America. The aims are clear and have been achieved. 

I feel it is worth to publish in Forests after including some paleo data. 

The authors predict the spread of fires and the disappearance of barriers to its spreading as a result of long-term droughts. One of the consequences of the decrease of water level are plant population shifts in wetland ecosystems e.g., the spread of the Sphagnum mosses, which was observed in the Boreal (Loisel, J., Yu, Z., 2013) and Arctic Alaska (Galka et al., 2018).

In my opinion, the authors of this study should include the results and observation from palaeoecological data for introduction and discussion chapter. 

The resistance to fire and regeneration after fire of plant populations on peatlands would be particularly important. It is common knowledge that fire is a threat but also an opportunity for some ecosystems.

The authors may take a deeper look at papers e.g. Kuhry 1994; Magnan et al., 2012; Camil et al., 2009 and papers published by Lukenbach et al., 2015, Journal of Hydrology 530: 405-418 and J Hydrology 548: 741-753.

>>> Thank you for your comment.  We added the following at the end of the first paragraph in the discussion: “Even in the absence of fire, North American peatlands under climate change have been shown via paleoecological analyses to be shifting away from less-flammable open and treed fens and towards enhanced Sphagnum moss cover [Loisel and Yu, 2013] that may ultimately lead to a greater abundance of more-flammable treed and forested bogs.  While the ground vegetation community of boreal peatlands is largely resilient to wildfire disturbance [Magnan et al., 2012], frequent, high severity fire decreases the abundance of the deciduous Larix laricina in treed and forested rich fens [Busque and Arsenault, 2005], which serves to dampen fire spread through high foliar moisture content [Schiks et al., 2016].”

>>We believe this additional text addresses the important climate change feedbacks related to fire in peatlands, where climate change impacts flammability (in the absence of fire), and fire itself may in some cases enhance long-term flammability by suppressing Larix growth

References

Loisel, J., Yu, Z., 2013. Recent acceleration of carbon accumulation in a Boreal peatland,

south central Alaska. J. Geophys. Res. 118, 41–53.

Galka, M., Swindles, G. T., Szal, M., Fulweber, R., Feurdean, A., 2018. Response of plant communities to climate change during the late Holocene: Paleoecological insights from peatlands in the Alaskan Arctic. Ecological Indicators, 85, 525–536

Kuhry P., 1994. The role of fire in the development of Sphagnum dominated peatlands in western boreal Canada. Journal of Ecology 82: 899–910.

Magnan GM, Lavoie M and Payette S., 2012. Impact of fire on long-term vegetation dynamics of ombrotrophic peatlands in northwestern Québec, Canada. Quaternary Research 77: 110–121.

Camill, P., Barry, A., Williams, E., Andreassi, C., Limmer, J., Solick, D.: Climate-vegetation-fire interactions and their impacton  long-term  carbon  dynamics  in  a  boreal  peatland  landscapein northern Manitoba, Canada, J. Geophys. Res., 114, G040.

#############

Reviewer 2 Report

Overview

This study sought to explore the role of peatlands in affecting fire spread and extent in the boreal plains of Canada under different drought regimes. They used GIS layers and Monte-Carlo simulations to simulate the fire regime. They found that drought increased fire extent in peatlands and the entire landscape.

 This paper highlights the fact that peatlands are considered a barrier to fire spread, but with increasing drought may indeed carry fire. This seems intuitive, and the analysis seems a bit surficial since the spatial patterns are not explored.  For example, why do some areas burn more than other under the drought regimes? Also, landscape connectivity could have been computed with Fragstats or LandscapeMetrics. Finally, tipping points are mentioned in the objectives but the analysis just focused on fire without explicitly looking at tipping points. That said, the paper was well-written and the analysis was sound.

Editorial comments 

Abstract

L13. I wouldn’t consider burning in peatlands to be a “driver”.

The Intro could be shortened to two sentences and still be adequate.

L27. I’m not sure why it’s disproportionate. 

Introduction

L71. It’s not clear to me how the transitions are based on thresholds that “increase fire growth rates”. Maybe just based on hydrological thresholds during drought?

L71. Again I don’t like the term “fire growth” because I’m not sure what you mean. Fire spread?

L72. Is landscape fire likelihood the same as fire risk?

L72. I don’t think Boreal Plains should be capitalized.

L75. Do you mean fuel continuity by landscape connectivity?

Methods

L79.  It would be nice to know the size of the study area. It's huge!

Figure 1.  It is hard to see the colors in the map.  You might need to have fewer categories so we can see them. Or zoom into a portion of the landscape to see the heterogeneity. Also there’s an odd hexagonal grid in the bottom right corner. Why not make the water blue? Also the area in the inset map doesn’t appear to match the shape of the larger map.

L100.  Shorten to “from 1980 through to 2017.”

L101. I think you mean “lightening ignitions cause 66% of the annual fires in the area.”

L110. The fuel types are additionally split into those types?  But those are broad categories. 

L113. This seems like an objective of your study.

L116-124.  This is very descriptive and might belong in the Intro or Discussion.

Table 1. Needs reformatting. Open appears in 2 rows.  Shouldn’t the % of study row should sum to 37%?

L253. How many consecutive days?

Results

L290-302. Instead of discussing slopes and intercepts, tell us what that means in ecological terms. It's not clear why it's important, as written.

Table 2. Aspen with phenology is an odd fuel type.  You’ll need to clarify how that differs from green aspen.

Figure 3. The moderate scenario wasn’t very moderate.  It appeared to be quite similar to the severe scenario and you might want to say more about why that was true in the discussion

Figure 5. I was surprised to see the effect of aspen > peat, which was not described in the discussion.

Discussion

L373. Dangling sentence.

L384-392. The idea that increasing drought and greater (dry) fuel continuity leads to larger fires seems like a given. A deeper dive into the spatial patterns based on fuel type etc. might be warranted. You mention severity, but only looked at extent. The severity story might be interesting as well.

Author Response

Overview

This study sought to explore the role of peatlands in affecting fire spread and extent in the boreal plains of Canada under different drought regimes. They used GIS layers and Monte-Carlo simulations to simulate the fire regime. They found that drought increased fire extent in peatlands and the entire landscape.

 This paper highlights the fact that peatlands are considered a barrier to fire spread, but with increasing drought may indeed carry fire. This seems intuitive, and the analysis seems a bit surficial since the spatial patterns are not explored.  For example, why do some areas burn more than other under the drought regimes? Also, landscape connectivity could have been computed with Fragstats or LandscapeMetrics. Finally, tipping points are mentioned in the objectives but the analysis just focused on fire without explicitly looking at tipping points. That said, the paper was well-written and the analysis was sound.

 >> We chose not to show calculations of FragStats in the paper (though an earlier draft had some FragStats computed for a fuel/non-fuel landscape in each drought scenario).  Instead, the model outcome of burn rate should be considered the landscape connectivity metric (see the last paragraph of the introduction where we state our objectives as burn rate = fuel continuity).

With regards to tipping points, we added a sentence in the discussion (line 430) to address this relevant comment: “Given the current modelling tools under Burn-P3, we are limited to three drought scenarios each with 100,000 fire simulations; future work may involve a more continuous treatment of landscape drought in order to determine key drying thresholds at large scales.”

Editorial comments 

Abstract

L13. I wouldn’t consider burning in peatlands to be a “driver”.

The Intro could be shortened to two sentences and still be adequate.

>> Removed the first and fourth sentence to shorten the first part of the abstract.

L27. I’m not sure why it’s disproportionate. 

>>> this was meant to relate that fire activity is anticipated to increased more in the peatland-rich areas, but removed for clarity.

Introduction

L71. It’s not clear to me how the transitions are based on thresholds that “increase fire growth rates”. Maybe just based on hydrological thresholds during drought?

>> simplified to “determines the hydrological thresholds in peatlands that correspond to markedly higher wildfire spread rates”

L71. Again I don’t like the term “fire growth” because I’m not sure what you mean. Fire spread?

>> that’s a term used in the wildfire simulation field, but changed to “fire spread” throughout the manuscript, as that term is more generic

L72. Is landscape fire likelihood the same as fire risk?

>>> We added a sentence at the end of the introduction to clarify this important point: “We do not consider the spatial patterns in human or natural values across the landscape that in turn impact fire risk (defined as the intersection of likelihood and effects [15]).” And added a citation to Finney 2005 in his authoritative work on the matter.

L72. I don’t think Boreal Plains should be capitalized.

>>corrected.

L75. Do you mean fuel continuity by landscape connectivity?

>> indeed we do.  Thanks for the suggestion to simplify that wording.

Methods

L79.  It would be nice to know the size of the study area. It's huge!

>> added at start of section 2.1  Area is 5.1 Mha.

Figure 1.  It is hard to see the colors in the map.  You might need to have fewer categories so we can see them. Or zoom into a portion of the landscape to see the heterogeneity. Also there’s an odd hexagonal grid in the bottom right corner. Why not make the water blue? Also the area in the inset map doesn’t appear to match the shape of the larger map.

>> blue water does not contrast as well with the greens used in the very abundant bog peatland type.  The grey is a better visual contrast.  The hex grid is shown as white lines over the water areas only, such as the large lakes of the southeastern corner of the study region.  The bright green area of the inset is the study area, the dark grey areas are the entire boreal and taiga plains ecoregions. We added to the caption of figure 1: “The 10 kha hexagon grid is shown over water only.”

L100.  Shorten to “from 1980 through to 2017.”

>> shortened as suggested

L101. I think you mean “lightening ignitions cause 66% of the annual fires in the area.”

>> changed as suggested

L110. The fuel types are additionally split into those types?  But those are broad categories. 

>> clarified to: “…predict the fire rate of spread in one of 16 fuel types, which are fall within one of the following classes”.  That is, there are 17 types, specifically 7 conifer, 1 open, 2 deciduous, 4 mixedwood and 3 slash.  Updated the 16 number to 17 to reflect the recent changes to the FBP system.

L113. This seems like an objective of your study.

>> It is unclear what you are referring to here.  We outline the objectives of the study at the end of the introduction.

L116-124.  This is very descriptive and might belong in the Intro or Discussion.

>> We see the details of the FBP being most relevant in the methods section, so as to not dive too far down into the details of the fire behaviour models in the introduction itself.

Table 1. Needs reformatting. Open appears in 2 rows.  Shouldn’t the % of study row should sum to 37%?

>> we used the rounded 37% in the text, though the number is precisely 37.2% (with a fair amount of uncertainty in those estimates).  Fixed table formatting

L253. How many consecutive days?

>> Unclear to what you are referring to.

Results

L290-302. Instead of discussing slopes and intercepts, tell us what that means in ecological terms. It's not clear why it's important, as written.

>>> inserted a few sentences in the Results section to clarify this where it is more relevant to understanding the latter results (would get lost in the discussion): “Ecologically, this implies that all peatland sites had essentially the same rate of water table decline per unit increase in Drought Code.  Open (treeless) peatlands showed a consistently high water table across all bogs and fens; the treeless status of these peatlands is likely owing to their hydrological state of more frequent surface saturation.  Fens (both rich and poor) have a significantly higher water table given the same Drought Code value on the landscape, again an indicator of increased groundwater inputs in fens [35] and the requirement for drier conditions in fens before critically dry conditions for fire spread are reached.”

Table 2. Aspen with phenology is an odd fuel type.  You’ll need to clarify how that differs from green aspen.

>> Added a footnote to Table 2: “Aspen with phenology refers to a fuel type that is leafless (D-1) in spring and transitions to leaf-on (D-2) in summer.  This is in contrast with the above fuel types that remain as D-1 or D-2 throughout the entire year.”

Figure 3. The moderate scenario wasn’t very moderate.  It appeared to be quite similar to the severe scenario and you might want to say more about why that was true in the discussion

>> good point, as explained around line 200 and in Figure A2, the drought scenarios are based on Drought Code climatology and expected return intervals, and are not related to fire activity directly.

Figure 5. I was surprised to see the effect of aspen > peat, which was not described in the discussion.

 >>> good point.  We added the following to the end of the second paragraph in the discussion: “Aspen and other deciduous forest is often considered alongside peatlands as a part of the boreal forest landscape less capable of carrying fire [43].  Since aspen and deciduous forest generally has far less surface organic soil available for combustion [18,29] and is generally disconnected from groundwater systems sensitive to drought [15], no such drought-sensitive fuel type changes were simulated in these upland forests.  Even with this static representation of aspen forest flammability in the Burn-P3 model, a marked increase in the burn rate is observed in the moderate drought scenario for aspen-dominated forests (Figure 5).  This increase in non-peatland burn rate suggests a landscape contagion where increasing burning of peatlands impinges on adjacent aspen forests, especially where the two ecosystems intersect more frequently in the southern half of the study region (Figure A5) and where the largest increases in burn rate are observed (Figure 3).”

Discussion

L373. Dangling sentence.

>> corrected to “…. Impediment to wildland fire spread”

L384-392. The idea that increasing drought and greater (dry) fuel continuity leads to larger fires seems like a given. A deeper dive into the spatial patterns based on fuel type etc. might be warranted. You mention severity, but only looked at extent. The severity story might be interesting as well.

>> we mention severity in the last paragraph of the discussion “Larger fires tend to have a higher proportion of area burned at high severity, as well as a greater total area burned severely [57,58], with potential implications for post-fire regeneration of vegetation communities [13,59].” But severity modelling is not natively incorporated into the Burn-P3 model we use here.